# How investment in children shape fertility choices of families: Evidence from Pakistan

Olan Naz[1], Nayab[1], Muhammad Ibrahim[1], Ayesha Khan[2], Adnan Ahmad Khan[1]*

**1** Research and Development Solutions, Islamabad, Pakistan, **2** The Akhter Hameed Khan Foundation, Islamabad, Pakistan

* adnan@resdev.org

**Data Availability Statement:** We are using secondary data which is publicly available on the

## Abstract

Fertility patterns are transitioning globally in that couples are choosing to have fewer children as they become more affluent, and parents are investing more time and resources in the lives of their children than they can receive back. This change is more established in developed countries and is now being recognized in developing countries. We explored this phenomenon and its implications for family planning in Pakistan, hypothesizing a quantity-quality transition underway. We examine the correlation between increased investment in children's education and the use of family planning services among Pakistani families. We conducted a secondary analysis of publicly available data from the Pakistan Social and Living Standards Measurement (PSLM) survey and its complementary Household Integrated Economic surveys (HIES) for 2018–19 accessed through the Pakistan Bureau of Statistics (PBS) website. The study included married women of reproductive age (MWRA) aged between 15 to 49 years of age. The current use of different contraceptive methods by MWRA was the outcome variable, whereas the education expenditure per child, and mother-level, child-level, and household-level factors, as control variables. The study employed multinomial logistic regression to assess the correlation between contraceptive use and education expenditure per child while controlling for other variables using STATA (version 17.0, STATA Corporation, College Station, Texas, USA). Data from 24,024 MWRA and 56,128 children were analyzed. 7,584 (30%) households have no children while 1,658 (10%) don't send any child to school. All households that send children to school incur at least some education related expense. The rise in education spending outpaces rates of rise in household incomes, suggesting that education is procured as a luxury good. However, the rate of spending on education falls off from the third child onwards. After controlling for confounders, the odds of using contraceptives increases as education expenditure rise, from 1.172 [CI, 1.029,1.336] when they spend under PKR 2000 (USD 13) to 1.495 [CI, 1.327,1.683] if they spend more than PKR 13,000 (USD 84) annually on the education of a child, compared to no education expenditure at all. There is one exception in the case of households in the wealthiest quintiles located in rural areas, where FP use reduces. Our findings support the hypothesis of a quantity-quality transition in Pakistan, where increased wealth and educational investment in children are linked to reduced fertility and higher contraceptive use households. The use of FP increases from the poor to the richest wealth quintiles nationally and correlates with spending on the education of an older child. However, an

Pakistan Bureau of Statistics (PBS) website. Following is the link for the data: https://www.pbs.gov.pk/content/pslm-hies-2018-19-microdata

**Funding:** This work has been funded by the Bill & Melinda Gates Foundation (updated grant number INV-051108). The funding was received by the institution (Research and Development Solutions) and not directly by any of the authors. The funding agency had no role in study design, data collection and analysis, decision to publish, or preparation of the manuscript.

important exception was observed among the wealthiest rural households, where family planning use decreases despite higher income levels. This suggests that affluent women, particularly in rural areas, may opt for larger families due to limited labor market opportunities or cultural preferences. We describe a major social change that reflects evolving values in families.

## Introduction

Global fertility is declining, including in developing countries that are now starting to match Europe and North America in declining fertility (FP2030 Annual Progress Report 2022). As these countries progress in making family planning (FP) accessible, allowing couples to choose to have the families that they want, it will become increasingly important to understand what factors drive the demand for family planning in households in these countries.

Traditionally, children were considered productive assets of the family, and increasing affluence was associated with more children [1]. Early and frequent mortality of children led parents to have more children, expecting some would not survive to adulthood. This pattern began reversing around the 1830s and by the 1860s there was a clear correlation between rising affluence and fewer children in developed countries [2]. The trend has accelerated and is now evident in developing countries as well. This decline in fertility has been further facilitated by improvements in maternal and child healthcare. Innovations such as better access to healthcare services, access to safer childbirth practices, early diagnosis of complications, and improved medical technologies have significantly reduced maternal and child mortality rates. As survival rates for both mothers and children improve, families tend to opt for fewer children, focusing more on quality of life and long-term investments in their future [3, 4].Many theories have been presented to explain this transition. Becker suggested that the reasons for having children have changed. Rather than being viewed as productive assets, children are now valued for themselves, leading to a quantity-to-quality tradeoff [5–9]. This corresponds to the observations that in modern developed society, the financial returns that parents receive from their grown children in later life are far less than the time and investment they provide during their upbringing [10]. As the number of children per household falls, the investments in bringing up each child have increased, indicating that parents are having fewer children, on whom they are spending more time and resources resulting in better outcomes in terms of education and human capital development [11, 12]. Moreover, reducing fertility rates can lead to improvements in maternal and child health, educational attainment, and economic productivity, highlighting the positive outcomes associated with investing in fewer children but with higher quality care and education [13]. Furthermore, as investment in the lives of children increases when households become richer, the marginal change in spending on children outstrips the rate by which the income of the family increases, suggesting that spending on children follows the pattern of a "luxury good". In economics, a luxury good is something that is procured more when one becomes affluent and the rate of spending on the luxury good outpaces the rate of rise in one's income [14]. Parental education, technology, and the role of the state as a guarantor of support for the elderly may also have shaped these social mores [15–17].

While these dynamics are well understood in developed countries, they are also visible in developing countries. However, the blend of both the old and new paradigms is visible in individual locations and local contexts [18–21]. In rural areas, fertility remains high in many developing nations, including in South Asia, although the quantity-quality tradeoff is emerging.

Giannelli and Francavilla observed that entering the paid workforce was associated with higher use of contraception and fewer children among Indian mothers [22, 23]. In effect, as was noted in the USA, the marginal cost of having the next child, from both expenses on children and the costs of foregone wages during pregnancy and child-rearing, increases as the number of children rises, and eventually limits further fertility [24].

Pakistan is an outlier in the region in terms of how its contraceptive use has lagged behind its regional neighbors despite considerable investments and nominal government support [25, 26]. On the other hand, recent supply-side programs have successfully increased the contraceptive prevalence rate (CPR) only to have it fall once the programming stops, suggesting low internal demand for FP among households [27]. However, this hides considerable nuance. In reality, CPR has been nudging up, albeit slower than in the other regions [28]. Couples do make decisions to limit or space their fertility. Factors driving these decisions are not well understood and key drivers of FP use such as women's participation in the paid labor force [20, 22] which is 9% for urban women and 24% nationwide [29] remain too low in Pakistan to have a profound impact.

We hypothesize that a quantity-quality transition may already be underway in Pakistan. If so, it would manifest in a correlation between higher investment in children and the use of family planning. In the biennial Household Integrated Economic Survey [30] and its inter-linked Pakistan Social and Living Measurements Survey (PSLM) [31] education expenditure by the household is the only expense reported that exclusively focuses on children and was used as a proxy for investments in children. Using these data, we test the hypothesis investment in education of children of school going age correlates with the use of family planning among Pakistani families.

## Methodology

### Study design

We conducted a secondary analysis of publicly available data from the Pakistan Social and Living Standards Measurement (PSLM) survey and its complementary Household Integrated Economic Surveys (HIES) for 2018–19 accessed through the Pakistan Bureau of Statistics (PBS) official website.

### Setting and the survey

The surveys are conducted by the Pakistan Bureau of Statistics, with a sampling frame that includes all localities across Pakistan and is weighted according to the local populations. Modules on demographics, maternal and child health, household spending and healthcare utilization were used for this analysis. While separate surveys, in a given year, both PSLM and HIES are conducted on the same household. This allows comparison of social and economic indicators across the same household. It is noteworthy that HIES has been incorporated into the PSLM survey since 1999 and has been consistently maintained, resulting in a more comprehensive and streamlined approach to gathering socio-economic data.

There are in effect two separate surveys that alternate each year. The first is powered at the provincial level and has more in-depth questions in each module and includes under 25,000 interviews across Pakistan. This alternates with a second survey that is powered at the district level and asks fewer questions per module but interviews nearly 80,000 respondents. For our analysis, the more detailed provincial survey was used.

## Sampling frame and sampling technique

The survey uses census 2017 for sampling frame and cluster randomized sampling. The survey adopts a stratified two-stage sampling technique where the administrative divisions for all the provinces in urban areas and the administrative districts for all the provinces in rural areas are considered as independent stratums (PSLM 19).

## Participants

Married women of reproductive age (MWRA - aged between 15 to 49 years) were included in the study. The response rate for the study was 95%.

## Variables used

The binary dependent (outcome) variable was the current use of contraception MWRA. The current contraception use is defined as the active utilization of a contraceptive method or device at the time of data collection. Within the PSLM survey, current contraception use encompasses both modern and traditional methods. Modern methods include intrauterine devices (IUDs), injections, implants, female sterilization, male sterilization, pills, and condoms. Traditional methods comprise withdrawal, rhythm, and any other traditional methods reported by participants. Independent variables (Education expenditure per child represents the average amount spent on the education of all children within a family who are between the ages of 4 to 25 and are currently enrolled in school. Mother's employment status indicates the current employment situation of each woman, categorized as employed, unemployed, or not in the labor force/inactive. Mother's education level is classified into primary (1 to 5 years of education), middle (6–8 years of education), secondary (8–10 years of education), and higher education (above 10 years of education). The number of children refers to the total count of children born to each mother, ranging from no children to mothers with more than five children. The variable province and region specify the geographic location of the household within Pakistan, including regions such as Punjab, Sindh, Khyber Pakhtunkhwa, Balochistan, etc., categorized by rural or urban areas.) included 27 questions about education, occupation, family planning, household characteristics, household income and expenditure [30, 31]. Family expenditure on education per child [32, 33] is treated as the main independent variable while controlling for mother-level factors; mothers' education [34, 35], age, employment [36], and awareness of contraceptive methods, child-level factors; the number of children in a family (school going age; 4–25 years old) [37] and gender of the last child [36], household level factor; the household wealth index and demographics [38]; province and region.

## Data analysis

We used multinomial logistic regression to produce odds ratio (OR) and 95% confidence interval (CI).

We used the Demographic and Health Survey (DHS) index methodology developed by Filmer and Pritchett (1998) [39] to construct the wealth index using PSLM data. This approach involves assessing household wealth based on common asset ownership and household characteristics outlined in the questionnaire. Specifically, variables such as domestic servants, agricultural land, and house ownership. Subsequently, principal component analysis (PCA) was utilized to generate urban, rural and combined wealth scores, which are then used to classify households into wealth quintiles. Regression analysis is further employed to estimate composite scores for both rural and urban areas, with households evenly distributed across quintiles representing 20% in each category for rural, urban, and combined contexts.

Additionally, we assess the quality-quantity preference in contraceptive users and non-users by calculating the marginal effect. The marginal effect is calculated to comprehend and estimate the impact of the expected change in education expenditure. In addition, the study examines the variability in income brought on by increased wealth and its effects on education spending. The data was analyzed using STATA (version 17.0, STATA Corporation, College Station, Texas, USA).

## Results

A total of 24,024 families were surveyed with a mean of 2.23 (±2.10 SD) children per family. 7,584 (30%) have no children and 10,471 (42%) of families have 3 or more. 1,658 (10%) families do not send their children to school. The mean annual education expenditure for a family that spends on a child's education is PKR 34,883 (USD 227) annually and per child expenditure is PKR 14,329 (USD 93) annually (Based on 1 USD = 154 PKR, at the time of the survey). Further urban and rural segregation in education per child is PKR 22,106 (USD 144) and PKR 9,519 (USD 62) respectively. Only 17% of families spend more than PKR 13,000 (USD 84) on education per child, while 8.4% spend less than PKR 2,000 (USD 13). Among mothers, 60% have no formal education, and only 22% work (Table 1).

Spending on education increases initially from the first child to the second child but then falls for each subsequent child (Fig 1). In the overall context, the average age difference (spacing) among children is 7.82 years. However, this age gap is considerably shorter for children attending primary school (ages 5–10), with an average of about 2.84 years, and for those in middle school (ages 11–13), with an average spacing of approximately 0.75 years. Marginal change in education spending is the highest for the second child and then decreases sequentially for the third child and onwards.

In families with children of school going age, characterized by wider age gaps between siblings, the decreasing trend in education spending is less steep. Conversely, in families with closer spacing among children, the reduction in education spending for each additional child is notably steep.

The total number of children decreases as families become affluent. Furthermore, between each subsequent wealth quintile, the rate of increase in education spending outpaces the increase in total household income suggesting that education is procured as a luxury good (Column 4, Table 2). This may be accounted for by the fact that 55% of all children enrolled in higher education are from the wealthiest quintile.

Additionally, the mean income for households with educated members is PKR 390,727, significantly higher than that for households without educated members i.e. PKR 191,649 with a statistically significant difference and [-210,765.5, -187,392.1] on a 95% confidence interval, as indicated by a two-sample t-test.

Moreover, the data shows that educated households have significantly higher contraceptive use (CPR = 0.398) compared to uneducated households (CPR = 0.243), with a statistically significant difference and [-0.1714, -0.1383] on a 95% confidence interval. This further supports the hypothesis that families prioritizing education are more likely to invest in family planning. Furthermore, the CPR users have a significantly higher mean income of PKR 96,807 compared to non-CPR users, whose mean income is PKR 65,660. The difference of PKR 31,148 is statistically significant, with a 95% confidence interval of [−41,38, −21,258].

Education spending also correlates with higher CPR (Fig 2), which is nearly twice as high for families that spend on education, for any given number of children and rises with each child until a maximum of four children. Thereafter CPR falls but families that allocate

**Table 1.  Descriptive statistics of socio-demographic variables.**

| Variable | Percent |
|---|---|
| **Province** | |
| Punjab | 43.71 |
| Khyber Pakhtunkhwa | 21.52 |
| Sindh | 24.07 |
| Baluchistan | 10.7 |
| **Region** | |
| Rural | 65.14 |
| Urban | 34.86 |
| **Current Contraceptive use** | |
| No | 68.71 |
| Yes | 31.29 |
| **Education Expenditure (in PKR)** | |
| 0 | 45.66 |
| 1–2,000 | 8.44 |
| 2,000–3,600 | 8.86 |
| 3,600–6,500 | 9.7 |
| 6,500–1,3000 | 10.32 |
| >13,000 | 17.02 |
| **Number of Children** | |
| 0 | 30.16 |
| 1–2 | 28.2 |
| 3–5 | 33.76 |
| >5 | 7.88 |
| **MWRA's Age in years** | |
| 15–25 | 25.27 |
| 26–31 | 24.47 |
| 32–38 | 24.84 |
| 39–49 | 25.42 |
| **Mother's Education** | |
| No formal education | 60.64 |
| Primary | 12.56 |
| Middle | 7.14 |
| Secondary | 14.29 |
| Higher | 5.37 |
| **Mother's Employment Status** | |
| Not in the labor force | 77.67 |
| Employed | 21.85 |
| Unemployed | 0.48 |

resources for their school going children maintain a higher CPR compared to those who do not prioritize education spending.

Logistic regressions at the national level and separately for rural and urban areas further confirm this correlation while adjusting for wealth quintiles, number of children, mother's education, and urban residence. The odds of using contraceptives rise significantly with increasing spending on education, the number of children, whether the last child was a boy, the mother's education level, family wealth quintile, and urban residence. Mother's

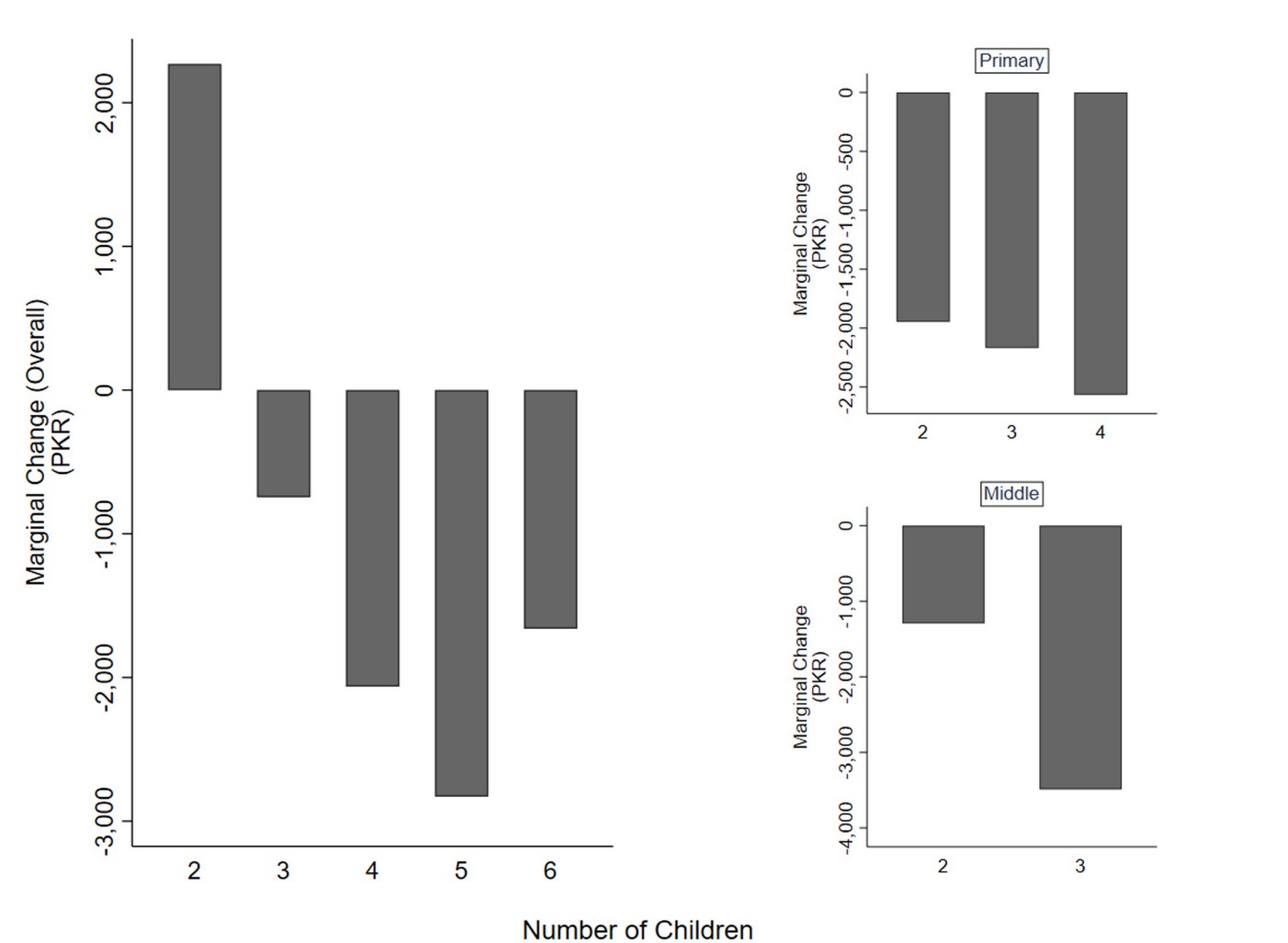

**Fig 1. Marginal change in education expenditure of children.** Note: The graph depicts the marginal change in mean education expenditure over the number of children for the overall, primary, and middle levels of education. For the primary and middle education graphs, only those families are included whose children are currently enrolled in school. The primary level consists of classes, one to five whereas the middle level consists of classes six to eight.

employment does not affect contraception use. These national level relationships are stronger at the urban level and slightly weaker at the rural level (Table 3).

Fig 3 shows that the probability of using contraceptives increases steadily and reaches its plateau when education expenditure reaches PKR 3,600, across all wealth quintiles in the national and rural sample but continues to rise in the urban sample. A similar relationship is seen with the increasing number of children.

## Discussion

We show that for over half of the population, a transition may be underway in which rising affluence is associated with lowering fertility in families. Thus, children may be desired for themselves rather than as productive assets for the family. Our findings also indicate that spending on education, a surrogate for investment in children, rises with affluence and at a higher rate than income growth, suggesting that education for children is procured as a luxury

**Table 2. Marginal change in annual income and education expenditure.**

| Wealth Quintiles | Number of Children (Mean) | Mean Household Income (PKR) | Marginal Change (%) in HH income | Proportion of Income to Education Expenditure (%) | Mean Household Education Expenditure (PKR) | Mean Education Expenditure Per Child (PKR) | Marginal Change in average expenditure per child (%) in (6) |
|---|---|---|---|---|---|---|---|
| | (1) | (2) | (3) | (4) | (5) | (6) | (7) |
| Poor | 4.5 | 251,602 | - | 3.3 | 8,268 | 1,808 | - |
| Lower-Middle | 4.4 | 293,861 | 42,259 (16.7) | 4.9 | 14,405 | 3,053 | 1,245 (68.1) |
| Middle | 4.2 | 362,941 | 69,080 (23.5) | 6.8 | 24,714 | 5,165 | 2,111 (69.1) |
| Upper Middle | 4.0 | 452,500 | 89,558 (24.7) | 9.0 | 40,821 | 8,739 | 3,573 (69.2) |
| Rich | 3.6 | 702,272 | 249,772 (55.1) | 13.3 | 93,476 | 21,261 | 12,522 (143.3) |

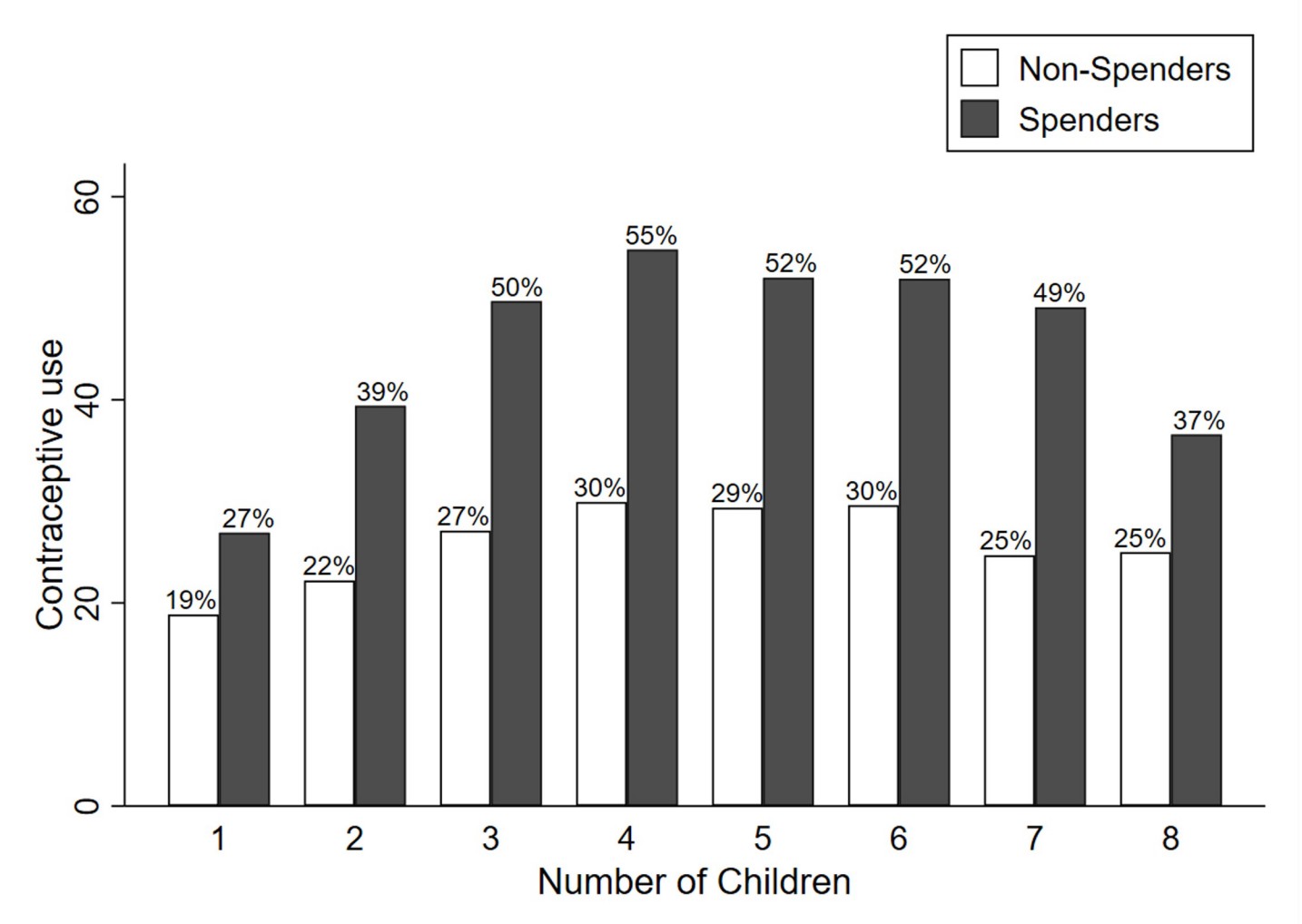

**Fig 2. Contraceptive use by education spenders and non-spenders and number of children.** Note: To test the statistical significance the Two-sample t-test was run. The result indicates that there is a statistically significant difference in mean contraceptive use of education spenders vs. non-spenders.

**Table 3. Effect of education expenditure on contraceptive use and non-use using logistic regression.**

| Dependent Variable: Contraceptive use | (I) National | (II) Urban | (III) Rural |
|---|---|---|---|
| **Education Expenditure in PKR [Base: 0]** | | | |
| 1–2000 | 1.172* | 1.185 | 1.149 |
| | [1.029,1.336] | [0.900,1.561] | [0.987,1.337] |
| 2000–3600 | 1.430* | 1.310* | 1.418* |
| | [1.261,1.621] | [1.034,1.658] | [1.220,1.648] |
| 3600–6500 | 1.277* | 1.265* | 1.251* |
| | [1.129,1.445] | [1.013,1.579] | [1.077,1.453] |
| 6500–13000 | 1.450* | 1.299* | 1.514* |
| | [1.282,1.639] | [1.064,1.586] | [1.295,1.770] |
| >13000 | 1.495* | 1.588* | 1.350* |
| | [1.327,1.683] | [1.324,1.905] | [1.147,1.589] |
| **Number of Children [Base: 0]** | | | |
| 1–2 | 1.491* | 1.528* | 1.503* |
| | [1.306,1.702] | [1.238,1.885] | [1.264,1.789] |
| 3–5 | 3.655* | 3.673* | 3.818* |
| | [3.140,4.253] | [2.880,4.685] | [3.135,4.649] |
| >5 | 4.378* | 4.456* | 4.487* |
| | [3.640,5.266] | [3.256,6.055] | [3.551,5.669] |
| **Mother's Age [Base: 15–25]** | | | |
| 26–31 | 1.104 | 1.091 | 1.103 |
| | [0.981,1.242] | [0.909,1.310] | [0.944,1.289] |
| 32–38 | 1.129 | 1.055 | 1.164 |
| | [0.993,1.284] | [0.862,1.291] | [0.984,1.377] |
| 39–49 | 1.026 | 0.796* | 1.189 |
| | [0.898,1.173] | [0.645,0.982] | [0.998,1.416] |
| **Mother's Employment Status [Base: Not in Labor Force]** | | | |
| Employed | 1.040 | 1.266* | 0.959 |
| | [0.960,1.127] | [1.085,1.477] | [0.872,1.054] |
| Unemployed | 0.706 | 1.327 | 0.445* |
| | [0.435,1.146] | [0.622,2.829] | [0.223,0.890] |
| **Mother's Education [Base: No Formal Education]** | | | |
| Primary | 1.290* | 1.171 | 1.380* |
| | [1.167,1.427] | [0.994,1.379] | [1.213,1.569] |
| Middle | 1.545* | 1.541* | 1.559* |
| | [1.359,1.758] | [1.282,1.851] | [1.299,1.870] |
| Secondary | 1.474* | 1.484* | 1.435* |
| | [1.325,1.641] | [1.274,1.729] | [1.231,1.674] |
| Higher | 1.506* | 1.430* | 1.576* |
| | [1.285,1.766] | [1.163,1.759] | [1.197,2.077] |
| **Wealth Quintiles [Base: Poor]** | | | |
| Lower Middle | 1.295* | 1.125 | 1.183* |
| | [1.162,1.443] | [0.956,1.323] | [1.029,1.361] |
| Middle | 1.543* | 1.254* | 1.483* |
| | [1.377,1.727] | [1.057,1.488] | [1.282,1.714] |
| Upper Middle | 1.559* | 1.265* | 1.673* |
| | [1.374,1.769] | [1.061,1.509] | [1.439,1.946] |

(*Continued*)

**Table 3.** (Continued)

| Dependent Variable: Contraceptive use | (I) National | (II) Urban | (III) Rural |
|---|---|---|---|
| Rich | 1.565* | 1.171 | 1.601* |
| | [1.344,1.823] | [0.947,1.448] | [1.354,1.894] |
| **Region [Base: Rural]** | | | |
| urban | 1.455* | - | - |
| | [1.338,1.583] | | |
| **Observations** | 20,035 | 7,029 | 13,006 |

Note: * represents significance at 5% level. Confidence intervals are reported in the square brackets. Additionally, each model has controlled for MWRA's awareness of contraceptive methods and province, not shown in the table.

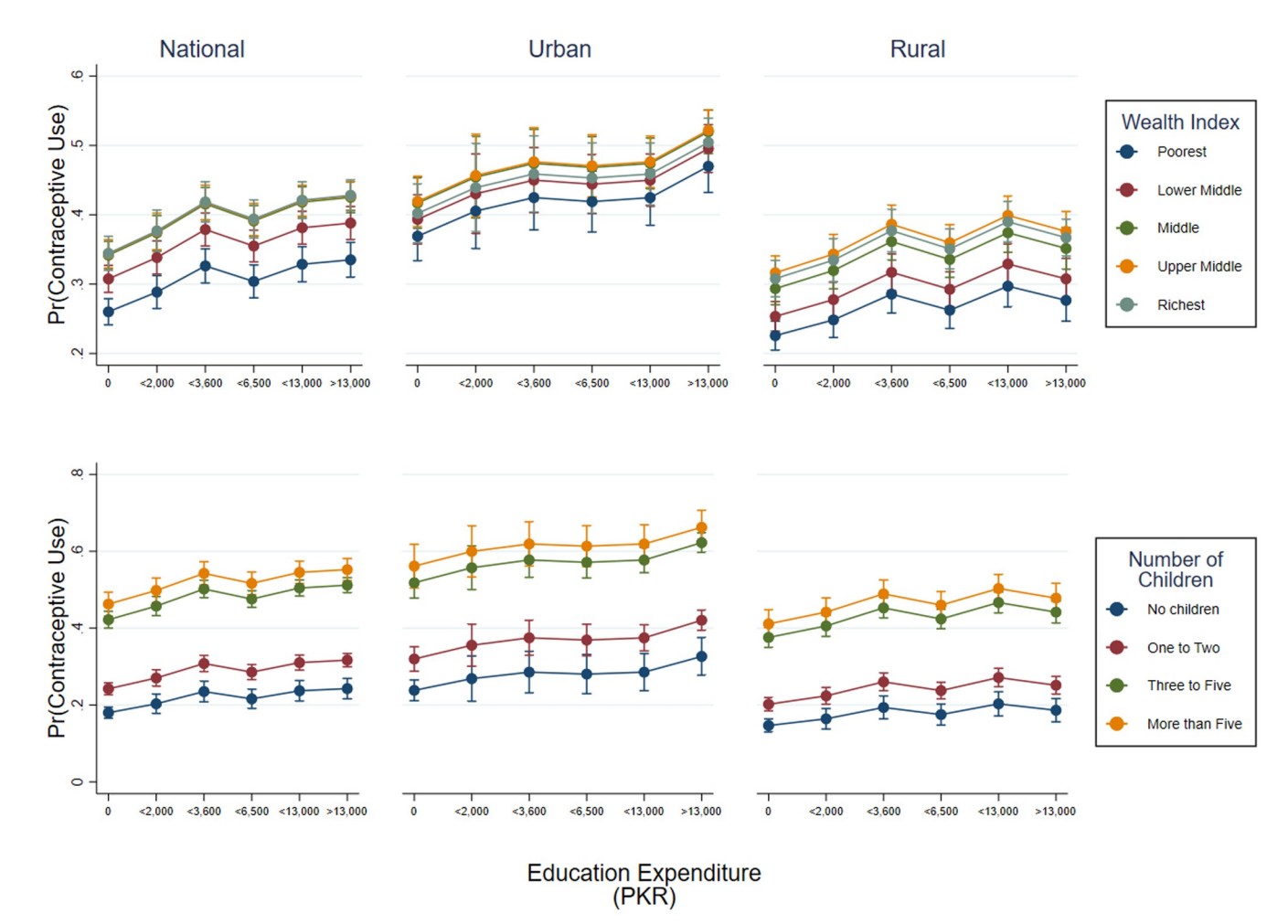

**Fig 3. Predictive margins of education expenditure and contraceptive use by wealth quintiles and number of children.** Note: The first row in the figure shows the margin plots of education expenditure and contraceptive use by wealth quintiles while the second row depicts the education expenditure and contraceptive use against the number of children in national, urban and rural settings.

good. On average, educated households have an annual income that is PKR 199,078 higher than uneducated households, and they exhibit a higher contraceptive prevalence rate (CPR) of 0.155. Additionally, increased spending on a child's education is associated with having fewer children and a CPR that is nearly twice as high for school going than non-school school going children.

Becker's theory of fertility posits that with increasing wealth, families value children for themselves (and therefore invest in their futures) rather than as productive assets [6, 8, 40]. This results in fewer children as families that grow richer, spend more time and resources with each child, in essence yielding a quality for quantity shift [41]. In fact, the resources parents invest in raising their children are far more than the potential future benefits they will receive from these children [10]. This has been posited as a major factor that led to a major decline in fertility in developed countries in the past two centuries [2, 42, 43]. Other factor include increased access to education and employment opportunities for women, changes in family structures, and the diffusion of new cultural norms regarding family planning and child-rearing practices [41]. We show that this phenomenon may well be underway in developing countries such as Pakistan, and elsewhere, albeit with some nuances [44].

Foremost, only around half of the households pay for a child's education. When they do, such families have rates of family planning that are twice as much as families that don't spend anything on education [45]. This may have implications for health workers seeking to counsel families in FP. Simply asking whether they are spending on education may actually help identify those who may be more open to the suggestion of FP use.

There is a more recent nuance in the fertility transition and affluence. Fertility has been nudging up for the most educated and rich mothers in developed countries in the past two decades [34, 46]. Doepke and others have posited that these rich, educated women when facing relatively lower wages than men for similar work, choose to stay home. These women tend to have more children than they would have if they had been working [46].

Our data shows aspects of this phenomenon in Pakistan. The probability of contraceptive use rises for the middle, upper middle, and richer quintiles and is bolstered further as education spending increases. This effect plateaus at around PKR 3,600 in education spending after which increasing education spending is not associated with contraceptive use. Women from the richest quintile have more children and use FP less, while their household expenditure on education continues to rise. This may be a reflection of what Doepke and colleagues describe for rich women from developed countries [34, 47, 48]. These affluent and educated Pakistani mothers may be finding that labor market options are not commensurate with their education or expectations and instead are staying home where in turn they are having more children. These findings suggest that policies aimed at enhancing women's education and employment opportunities may effectively reduce motivation for large families by highlighting the opportunity costs associated with childbearing, particularly for poor and middle quintiles and more equitable employment would be needed to attract more affluent and educated women [49, 50].

Such dividends may be more pronounced in the cities, where labor force participation for women is 9% compared to 24% overall [29]. Women who are employed, tend to have fewer children and a higher CPR. This suggests that women's employment creates both financial and opportunity costs associated with childbearing. Educated women, who are more likely to participate in the labor force, have greater income-earning potential as evident by our analysis, which likely incentivizes the use of family planning as they balance career aspirations with family life. Given that the effects of education and affluence affect FP use more in urban areas, any policies that enhance women entering workplaces in cities may change fertility rates far more than in rural areas.

We show that marginal spending on education per child peaks at the second child and falls for subsequent ones. It may be that relatively poor families run out of resources after having paid for the education of two children and therefore for poor families with five to six children only the first few children may receive an education [51]. Given the importance of education in the modern economy, this is a double strike for children from the poorest families.

## Limitations

While our study offers insights into the relationship between investment in children's education and fertility preferences, it is important to acknowledge the limitations. The survey data we utilized did not comprehensively capture investment in children, particularly in terms of health-related costs. These costs, including expenses related to healthcare services, and medical treatment, were not disaggregated for household members. As a result, our analysis may not fully account for the overall investment in children's well-being beyond education. Future research could incorporate data on health and nutrition expenditures to better understand how these factors contribute to the quality of children and influence fertility preferences.

## Conclusion

Our study highlights the complex global trends in fertility, both reduction in fertility with affluence and slight increase in fertility among the richest families, are also evident in Pakistan. These findings contribute to understanding the demand for family planning in Pakistan and provide important programmatic information to implementers, for example, in identifying households that may be more open to future use of family planning. Policies that enhance women's access to education and equitable employment opportunities could contribute to both economic growth and more widespread use of family planning. It also suggests that women's entry into paid income generation may be a crucial driver of family planning. Future research may further elaborate on these findings to understand the factors that further shape fertility choices in families.

## Author Contributions

**Conceptualization:** Muhammad Ibrahim, Adnan Ahmad Khan.

**Data curation:** Olan Naz, Muhammad Ibrahim.

**Formal analysis:** Olan Naz,  Nayab, Muhammad Ibrahim.

**Funding acquisition:** Adnan Ahmad Khan.

**Investigation:** Olan Naz.

**Methodology:** Olan Naz,  Nayab, Muhammad Ibrahim.

**Project administration:** Ayesha Khan.

**Resources:** Adnan Ahmad Khan.

**Software:** Olan Naz,  Nayab.

**Supervision:** Muhammad Ibrahim.

**Validation:**  Nayab, Muhammad Ibrahim, Ayesha Khan.

**Visualization:** Olan Naz,  Nayab, Muhammad Ibrahim.

**Writing – original draft:** Olan Naz.

**Writing – review & editing:** Olan Naz, Muhammad Ibrahim, Adnan Ahmad Khan.

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
