## [Decision Letter · Decision Letter 0]

21 Feb 2024

PGPH-D-23-02225

Do social preferences for children play a role in fertility preferences for the family: Evidence from Pakistan

Dear Dr. Khan,

Thank you for submitting your manuscript to PLOS Global Public Health. After careful consideration, we feel that it has merit but does not fully meet PLOS Global Public Health’s publication criteria as it currently stands. Therefore, we invite you to submit a revised version of the manuscript that addresses the points raised during the review process.

We appreciate this submission and its contribution to a better understanding of fertility and the use of contraception in Pakistan.

Please review and respond to the requests and suggestions from the reviewers. As Reviewer 1 noted, we suggest a broader literature review, as the correlation of increasing use of contraception with increasing wealth has been widely studied and documented. 

Please define all terms and variables used.

We also suggest that you expand on your conclusion to address how your findings can be of use in Pakistan and beyond.

In addition, please ensure that the manuscript is copy edited. In several instances, the phrasing of sentences is awkward. If possible, we suggest that you work with a copy editor on your revised manuscript. 

We look forward to receiving your revised manuscript.

Kind regards,

Laura Miniea Hoemeke, DrPH

Academic Editor

Journal Requirements:

1. Please amend your detailed online Financial Disclosure statement. This is published with the article. It must therefore be completed in full sentences and contain the exact wording you wish to be published.

a) State the initials, alongside each funding source, of each author to receive each grant. For example: "This work was supported by the National Institutes of Health (####### to AM; ###### to CJ) and the National Science Foundation (###### to AM)."

2. Please update your online Competing Interests statement. If you have no competing interests to declare, please state: “The authors have declared that no competing interests exist.”

3. Please provide separate figure files in .tif or .eps format only and remove any figures embedded in your manuscript file. Please also ensure that all files are under our size limit of 10MB. You may leave the figure captions or legends in the manuscript.

4. Please ensure that you refer to Table 1 in your text as, if accepted, production will need this reference to link the reader to the table.

Additional Editor Comments (if provided):

Reviewers' comments:

Reviewer's Responses to Questions

**Comments to the Author**

1. Does this manuscript meet PLOS Global Public Health’s publication criteria? Is the manuscript technically sound, and do the data support the conclusions? The manuscript must describe methodologically and ethically rigorous research with conclusions that are appropriately drawn based on the data presented.

Reviewer #1: No

Reviewer #2: Yes

2. Has the statistical analysis been performed appropriately and rigorously?

Reviewer #1: No

Reviewer #2: I don't know

3. Have the authors made all data underlying the findings in their manuscript fully available (please refer to the Data Availability Statement at the start of the manuscript PDF file)?

Reviewer #1: Yes

Reviewer #2: Yes

4. Is the manuscript presented in an intelligible fashion and written in standard English?

Reviewer #1: Yes

Reviewer #2: No

5. Review Comments to the Author

Reviewer #1: Naz et al. submitted a secondary data analysis on the role of education expenditure on contraception use in Pakistan. While in interesting research question, there are several major issues with the methods that unfortunately lead me to have to reject the submission. I have provided my comments below, specific to the methods, for the authors to consider:

1. I would urge the authors to conduct a broader literature review to discuss current research and how theirs contribute to the overall topic.

2.

3. I am concerned about the authors use of two separate surveys. There is no mention in the method if these two surveys can be combined (or previously have been). They state that they have been conducted on the same household. Does that mean that they both have the exact same sample size and participants? More information is needed as there are concerns about the validity.

4. In the objective, the authors state that they are looking at the education of older children. However, “older children” is never defined. I would urge the authors to define this population. Moreover, I would also recommend the authors to consider their study population as girls as 15 could be considered “older children”. Moreover, can a girl age 15 have an older child?

5. The authors need to more clearly define their participants. For example, were women who did not have children excluded from the sample? What about women who did not answer the outcome question? I would urge the authors to a flow diagram of how they stratified the sample.

6. Please define “current contraception use”. What types of contraception does it include?

7. There should only be one independent variable. From my understanding of the objective, it is education expenditure. This is the most concerning issue of the manuscript.

8. Please define all variables in the analysis.

9. Please provide some citations and evidence as to why certain variables were chosen as controls. I would urge the authors to create a DAG to demonstrate how the controls were determined. I am concerned about the validity of the regression given this lack of information.

10. Please note which variables were used in the wealth index and how it was calculated

11. Please provide more information on how the regression was developed and executed.

12. It is unclear what the quality-quantity preference of contraceptive users is. I would urge the authors to simplify their analysis to only the regression.

13. It is unclear why families with no children were included in this analysis. How can you have education expenditure if you do not have a child? Please provide a justification

14. The last child sex in Table 1 is unnecessary.

15. Please provide an x-axis label on all of the graphs

16. I am very concerned that mother’s age was not a considered variable in the regression as this would clearly impact her ability to have children.

17. Rather than categorizing expenditure, the authors should consider a linear regression

18. Please only report the outcome variable in the regression.

19. I would urge the authors to do a broader literature review for the discussion to compare and contrast their findings.

20. Please provide strengths and limitations

Reviewer #2: Overall the paper is clear, with the data and corresponding conclusions well aligned. The subject is interesting, and the contributing to understanding fertility trends is useful. The analysis appears to be well conducted and rigorous (but as I am not an expert in statistics I cannot comment completely on the appropriate and correct use of analysis techniques).

The manuscript could benefit from some minor revisions as follows, however.

Writing:

The writing is intelligible, but there are some instances where the language is a little ambiguous and a few instances of grammatical or English language errors. This is only a matter of copyediting however, which I would encourage the authors to review by engaging a native English speaker on their review. Some examples of this include (but are not limited to):

- line 268-269: 'These women tend to have more children than they would have if they been working', which needs correcting to 'These women tend to have more children than they would have if they had been working'

- line 277: 'These affluent and educated Pakistani mothers may be finding that labor market options are not commiserate with their education or expectations...' which needs correcting to 'These affluent and educated Pakistani mothers may be finding that labor market options are not commensurate with their education or expectations...'

Tables and Figures:

The tables and figures are very useful, but would benefit from better labelling for improved interpretability. For example:

- Figure 1 needs a label on the X axis, and a reference to the unit of measurement on the Y axis

- Figure 2 needs a label on the X axis, and a reference to the unit of measurement on the Y axis (assume it is CPR, but this isn't fully clear)

- Table 3 of the regression outputs is quite hard to interpret for an everyday reader. This could be moved to an annexe.

- Figure 3 needs units on the X axis label, and clearer/more visible labels (they're a bit hard to see).

Other points:

- It would be helpful if the research's main hypothesis / research question were more prominent in the abstract. I would suggest adding this to the last line of the introduction in the abstract. This would really help with clarity.

- Similarly the title is a little ambiguous and could better reflect the hypothesis of the research. I would suggest it more specifically reflect the research question being investigated, for example 'Does investment in children('s education) play a role in fertility preferences in Pakistan?', or 'Does investment in children have a bearing on family size and contraceptive use in Pakistan?' etc. Currently it's not very clear from the title what the article is about.

- The policy implications of the findings are not well articulated, especially for the wealthiest quintiles (which diverges from the overall trend). It would be interesting to compare with other contexts how these phenomena have influenced policy and decision making elsewhere.

- Additionally, the point that increased CPR correlates with the sex of the last child being male is interesting, and I think warrants being acknowledged further than one line. It would be interesting to hear the authors' policy recommendations related to this.

- Lastly, the research could benefit from a limitations section. This could include, for example, that the data used only surveys married women; two surveys are used, albeit of the same households, but the data from each cannot be matched to specific households; and that the data is only indicating correlation, and what this means for its interpretation (etc).

6. PLOS authors have the option to publish the peer review history of their article (what does this mean?). If published, this will include your full peer review and any attached files.

**Do you want your identity to be public for this peer review?** For information about this choice, including consent withdrawal, please see our Privacy Policy.

Reviewer #1: No

Reviewer #2: No

---

## [Decision Letter · Decision Letter 1]

1 Oct 2024

PGPH-D-23-02225R1

How Investment in Children Shape Fertility Choices of Families: Evidence from Pakistan

Dear Dr. Khan,

Thank you for submitting your manuscript to PLOS Global Public Health. After careful consideration, we feel that it has merit but does not fully meet PLOS Global Public Health’s publication criteria as it currently stands. Therefore, we invite you to submit a revised version of the manuscript that addresses the points raised during the review process.

We look forward to receiving your revised manuscript.

Kind regards,

Laura Miniea Hoemeke, DrPH

Academic Editor

Journal Requirements:

Additional Editor Comments (if provided):

Thank you for submitting this revised manuscript.

As noted previously, your manuscript addresses an important topic. Thank you for addressing questions and comments from reviewers.

Please note suggestions from Reviewer #3.

In addition, we suggest that you provide further analysis of the theory of fertility that "posits that with increasing wealth, families value children for themselves (and therefore invest in their futures) rather than as productive assets" in light of changing lifestyles and primary sources of income in Pakistan. The possibility should be considered that families are willing to pay for education to improve long-term income-earning potential of children as a long-term investment in the families' income and wealth. In addition, as noted by Reviewer #3, income-earning potential of women in the labor market should be addressed. The discussion and conclusion section of your manuscript would be strengthened by adding this additional analysis.

In addition, before submitting your revised manuscript, we suggest that you work with a copyeditor to improve the overall readability of your manuscript.

Reviewers' comments:

Reviewer's Responses to Questions

**Comments to the Author**

1. If the authors have adequately addressed your comments raised in a previous round of review and you feel that this manuscript is now acceptable for publication, you may indicate that here to bypass the “Comments to the Author” section, enter your conflict of interest statement in the “Confidential to Editor” section, and submit your "Accept" recommendation.

Reviewer #3: All comments have been addressed

Reviewer #4: (No Response)

2. Does this manuscript meet PLOS Global Public Health’s publication criteria? Is the manuscript technically sound, and do the data support the conclusions? The manuscript must describe methodologically and ethically rigorous research with conclusions that are appropriately drawn based on the data presented.

Reviewer #3: Yes

Reviewer #4: Yes

3. Has the statistical analysis been performed appropriately and rigorously?

Reviewer #3: Yes

Reviewer #4: Yes

4. Have the authors made all data underlying the findings in their manuscript fully available (please refer to the Data Availability Statement at the start of the manuscript PDF file)?

Reviewer #3: Yes

Reviewer #4: Yes

5. Is the manuscript presented in an intelligible fashion and written in standard English?

Reviewer #3: Yes

Reviewer #4: Yes

6. Review Comments to the Author

Reviewer #3: Clarify Hypothesis and Objective: Explicitly state the hypothesis and main objective in the abstract to make it clear from the outset what the study aims to demonstrate.

Highlight Key Findings: In the conclusion, briefly emphasize the most significant findings and their implications for family planning policies, especially focusing on the observed exceptions in wealthier rural households.

Methodology

Clear and Detailed: The methodology is clearly described and well-structured, providing necessary details on the data sources, sampling, and analysis.

Comprehensive Analysis: Appropriate use of multinomial logistic regression and control for various factors enhances the study's reliability.

Discussion

strong Theory Integration: Effectively ties findings to Becker's theory of fertility and global trends.

Policy Relevance: Highlights key policy implications for education and employment opportunities for women.

Reviewer #4: It was a pleasure reviewing this insightful manuscript (Do social preferences for children play a role in fertility preferences for the family: Evidence from Pakistan). I only have minor comments for authors consideration.

Abstract:

Methodology: How did study authors get access to the data used for this study

Introduction:

- Lines 82-85: I would recommend authors consider the influence of innovation in terms of improvement in maternal and child health and impact on child and mother mortality

- Lines 91-94: I would expect that as the number of children decreases, investment of children and a direct increase on reserves (finance). The sentence at its current state isn’t clear

Methodology:

- Setting and the Survey: How did authors access this data for analyses?

- Authors should consider providing more details on how data was abstracted from the database.

- Data analysis: How were the characteristics of study participants gotten?

Was the data set weighted after extracted from the database? I have seen how this was done.

7. PLOS authors have the option to publish the peer review history of their article (what does this mean?). If published, this will include your full peer review and any attached files.

**Do you want your identity to be public for this peer review?** For information about this choice, including consent withdrawal, please see our Privacy Policy.

Reviewer #3: **Yes: **Sarashwati Giri

Reviewer #4: No

---

## [Editor Report · Decision Letter 2]

29 Oct 2024

How Investment in Children Shape Fertility Choices of Families: Evidence from Pakistan

PGPH-D-23-02225R2

Dear Dr Khan,

We are pleased to inform you that your manuscript 'How Investment in Children Shape Fertility Choices of Families: Evidence from Pakistan' has been provisionally accepted for publication in PLOS Global Public Health.

Best regards,

Laura Miniea Hoemeke, DrPH

Academic Editor